# Modeling and Simulation of the Casting Process with Skeletal Sand Mold

**DOI:** 10.3390/ma13071596

**Published:** 2020-03-31

**Authors:** Jinwu Kang, Jiwu Wang, Haolong Shangguan, Lele Zheng, Chengyang Deng, Yongyi Hu, Jihao Yi

**Affiliations:** 1School of Materials Science and Engineering, Key Laboratory for Advanced Materials Processing Technology, Tsinghua University, Beijing 100084, China; shangghl@163.com (H.S.); an123yi@126.com (C.D.); huyy@mail.tsinghua.edu.cn (Y.H.); 13881717307@163.com (J.Y.); 2School of Mechanical, Electronic and Control Engineering, Beijing Jiaotong University, Beijing 100044, China; jwwang@bjtu.edu.cn (J.W.); 17121257@bjtu.edu.cn (L.Z.)

**Keywords:** skeletal sand mold, casting, modeling and simulation, heat transfer, stress

## Abstract

The author-proposed skeletal sand mold, which mainly includes a shell, air cavities and a truss support structure, has been experimentally proven to be very useful in controlling the cooling of casting at local areas and at different periods of the casting process. The modeling and simulation of the casting process using a skeletal sand mold were systemically analyzed. Complicated casting/mold and mold/air boundaries, and the thermal and mechanical behavior of the skeletal sand mold during the casting process were highlighted. A numerical simulation of the casting process of a stress frame specimen using a skeletal sand mold was performed. The temperature, stress and displacement fields of the casting and skeletal sand mold were obtained and compared with those using a traditional sand mold. The simulated results were validated with experiments. Using the skeletal sand mold, the cooling rate of the casting can be greatly improved due to the significant heat release from mold surface to environment. The residual stress and deformation of the casting can be reduced because of the decreased stiffness of this kind of mold. Although the skeletal sand mold is susceptible to cracking, it can be avoided by filleting in the conjunctions and increasing the shell thickness.

## 1. Introduction

The key process of casting is the solidification of a liquid metal inside a mold regardless of the mold material and molding method. There are many methods to control the cooling of a casting in a mold. For example, in permanent mold casting and die casting, water cooling channels and oil heating channels have been popularly adopted in metallic dies. Norwood et al. [1] reviewed the effect of conformal cooling channel and traditional cooling channel designs. Karkkainen et al. [2] simulated the heat transfer of single-phase turbulent flow in a ‘bubbler’ geometry, which is typically used for cooling channels in a high-pressure die-casting process. Zhao et al. [3] proposed circulating air, water or even liquid nitrogen inside pipes buried in a mold during molding to quicken the cooling of certain areas of a casting in sand mold. Showman et al. [4] dramatically modified the thermal properties of sand molds and cores using a low-density alumina-silicate ceramic (LDASC) as an additive to make thin wall iron castings. Grassi et al. [5] proposed an ablation casting method using a water curtain to erode away the sand mold bonded by water-soluble binders to achieve rapid solidification of a casting from one side to another. Kang et al. [6] proposed the PSIRC (post-solidification intensive riser cooling) method to achieve fast and even cooling of heavy steel castings. However, these methods are limited to certain casting methods or alloys. 

The appearance of 3D printing technology has brought new applications and ideas to the foundry industry in prototype making, pattern making, sand mold and core preparation, and even casting replacement. In particular, the production of cores and molds by additive manufacturing is gaining great traction. Druschitz et al. [7] discussed using binder jetting technology to construct sand molds for complex and cellular cast structures and fused filament construction and vat photo polymerization to produce complex investment casting patterns. Casalino et al. [8] used selective laser sintering (SLS) to produce sand molds for aluminum castings. Crommert et al. [9] employed SLS to make silica sand and zircon sand molds, and excellent castings were made. Lu et al. [10] applied 3D printing technology to construct investment molds for turbine blades, and the quality of the casting was improved and the cost was decreased. Compared with traditional manufacturing and forming methods, Snelling et al. [11] cast cellular structures using 3D-printed complicated sand molds. Li et al. [12] proved that 3D printing makes the construction of more complicated structures possible. Sama et al. [13] used 3D printing to the optimization of the casting orientation, hybrid molding, nesting multiple cast parts within a single mold and fabrication of complex freeform structures. However, 3D printing technologies have mostly been applied to print traditionally solid molds and cores without modification of their own structure. Shangguan et al. [14] presented an additive manufacturing-driven mold design for castings, the skeletal sand mold design, which mainly includes a shell layer, functional cavities and a lattice support structure. The design principles were proposed, and the application of these molds on an A356 Al alloy specimen casting was experimentally investigated. The results showed that the cooling of the casting was greatly improved compared with the usage of traditional sand molds. Furthermore, it provides the potential to further control the whole cooling process by applying cooling nozzles to interested local areas of the casting. Meanwhile, the weight of the sand mold was greatly reduced, which ultimately led to great savings of resin, printing time and 3D printing costs. Shangguan et al. [15] proposed a rib-enforced shell sand mold design, which greatly reduced the weight of the mold. Deng et al. [16] designed air cavities in the sand shell to enhance the insulation effect for risers. The new mold design brings a challenge to the modeling and simulation of the casting process for the traditional mold design.

In the present study, the modeling and simulation of the casting process based on the skeletal sand mold were investigated in a broad view, which would serve a guidance for the research of modeling and simulation corresponding to this kind of new mold design. Then, a case study about the simulation of a typical stress frame casting formed in a skeletal sand mold was analyzed to unveil their features of heat transfer, stress evolution and deformation.

## 2. Skeletal Mold Structure

A typical skeletal sand mold design is shown in Figure 1, and it includes a shell, functional structures and a lattice support structure. The shell structure which can be of a thickness variation corresponding to different locations of a casting gives the mold good heat dissipation capability. The support structure can make the mold self-supportive, and the functional structures such as air pockets, channels and chimney can achieve cooling, heating or insulation functions at interested locations. An air pocket is a kind of isolated air cavity in the shell of a sand mold, i.e., the shell is hollow somewhere. This unique structure of the skeletal sand mold brings complicated heat transfer features among the casting, sand mold and environment, and new resistance features of the sand mold affecting the contraction or expansion of a casting during the casting process. The modeling and simulation of the casting/skeletal sand mold system during the casting process face new challenges.

## 3. Modeling and Simulation of the Casting Process with the Skeletal Mold

### 3.1. Heat Transfer Analysis

#### 3.1.1. Basic Heat Transfer

The heat transfer in the casting and skeletal sand mold during the casting process mainly involves the conduction inside them. Heat conduction is governed by the Fourier equation.
(1)$$\rho c\frac{\partial T}{\partial t}=\frac{\partial }{\partial x}\left(k\frac{\partial T}{\partial x}\right)+\frac{\partial }{\partial y}\left(k\frac{\partial T}{\partial y}\right)+\frac{\partial }{\partial z}\left(k\frac{\partial T}{\partial z}\right)+\rho L\frac{\partial f_{s}}{\partial t}$$
where *ρ* is density, *c* is specific heat, *T* is temperature, *t* is time, *k* is thermal conductivity, *L* is the latent heat in unit volume of the melt or the heat release of resin burning of the sand mold and *fs* is the solid fraction of the melt or the burned fraction of resin in the sand mold. This equation can be used for both of the casting and sand mold.

#### 3.1.2. Burning of Resin in the Sand Mold

The shell of the skeletal sand mold is heated to a higher temperature than the traditional sand mold because of its low heat absorption capability. Therefore, the resin in the surface layer of the sand mold in contact with the casting will be burnt off during the casting process, then, heat will be released. Meanwhile, the thermal conductivity of sand mold maybe modified. Toth et al. [17] investigated the degradation behavior of the furan and the phenolic systems in 3D printed sand cores and found that the furan binder with quartz sand absorbed approximately 30% more heat compared to the phenolic system with cerabeads.

#### 3.1.3. Heat Transfer Boundary Conditions

1. Casting/Skeletal Sand Mold Interface

The heat transfer between the casting and sand mold is critical to the cooling of a casting. However, it is also very complicated, involving the evolution of an air gap and the contact status of the casting and mold. Xu et al. [18] investigated the heat transfer between a casting and sand mold by considering their deformation and contact status from the stress analysis results. However, for the skeletal sand mold, its shell temperature and resistance to the deformation of the casting are different from the traditional sand mold; thus, the casting/mold boundary needs to be further investigated.

2. Skeletal sand mold/environment boundary condition varying with location and time

Using the skeletal sand mold, the cooling of a casting can be controlled at any place, any time and any cooling rate through external forced cooling or even heating measures, such as water spray, forced air flow or resistance wire heating. These lead to complicated boundaries between the mold surface and air, sand wall and air pocket, cooling media and mold in the cooling channels and sand wall and chimney. Mainly, fluid flow, convection and radiation are involved. The irregular surface of the skeletal mold increases the area for radiation and convection. Meanwhile, because of its shell structure, its external surface temperature is higher than that of the traditional sand mold, thus, its convection and radiation heat transfers are more significant.

To achieve the cooling of a casting at any place and any time with a controlled cooling rate, numerous locations will be selected to set different boundary conditions. However, the skeletal sand mold surface is irregular and vast, including the external surface (which is exposed to air), the internal surface (which is exposed to the casting), and internal isolated surfaces (which form air pockets, air channels or chimneys). This kind of surface is different from the regular surface of a block-shaped mold. Thus, two new problems arise: the first is how to select and distinguish irregular surfaces, and the second is how to select numerous spots of interest for different cooling conditions.

The general convection and radiation boundary condition is
(2)$$k\frac{\partial T}{\partial n}=h(x,y,z,t)\left(T-T_{0}\right)-F_{v}\sigma \varepsilon \left(T^{4}-T_{0}^{4}\right)$$
where *n* is the normal direction of boundary, *σ* is the Boltzmann constant, *h* is the heat convection coefficient, *a* function of space and time, *ε* is the heat radiation emissivity and *Fv* is the view factor.

3. Convection

For natural convection of the mold surface to the environment, *h* can be taken as a constant or a function of mold surface temperature. 

For the natural convection in an air pocket, cooling channels and chimneys, the Nusselt number should be firstly calculated, and then the convection coefficient can be obtained. The convection coefficient *h* can be calculated by the following equation:(3)$$h\;=\;\frac{\lambda_{g}}{L^{*}}\cdot Nu$$
where *L** is the characteristic length of the air pocket, channels or chimneys, *Nu*, the corresponding Nusselt number. The calculation of *Nu* is different for natural convection and forced convection.

The Nusselt number in the convection in the channels by forced flow is calculated as follows:(4)Nu = C Ren⋅Prm
where *C* is a constant, *Re* is the Reynolds number, *Pr* is Prandtl number and Prandtl number is calculated by
*Pr* = *V*/*α*(5)
where *V* is the fluid flow velocity and α is its thermal diffusivity.
*α* = *λ_f_*/(*ρ_f_* · *cp_f_*)(6)
where *λ_f_, ρ_f_* and *cp_f_* are the thermal conductivity, density and specific heat of the fluid, respectively. 

The Reynolds number is calculated as
(7)Re = VL*ν
where *v* is a kinematic viscosity. 

Between the cooling pipe and the sand wall where there is significant temperature difference between them, Su [19] stated the Nusselt number calculation equation for turbulence flow.
(8)Nu =0.027 Re0.8⋅Pr1、3(μfμs)0.14
where *μ_f_* and *μ_s_* are the dynamic viscosity of the fluid at fluid temperature and sand wall temperature, respectively. 

For air flow, it is
(9)Nu =0.027 Re0.8⋅Pr1、3(TfTs)0.55
where *T_f_* and T_s_ are the air temperature and sand wall temperature, respectively. 

If the flow is stable, the Nusselt number is
(10)Nu =1.86 Re1/3⋅Pr1/3(dl)1/3(μfμs)0.14

For natural convection in a chimney, the Nusselt number is calculated by
(11)$$Nu\;=\;C{\left(Gr\cdot \mathrm{Pr}\right)}^{m}{\left(\frac{L^{*}}{H}\right)}^{n}=CRa^{m}{\left(\frac{L^{*}}{H}\right)}^{n}$$
where *H* is the height of the chimney, *Gr* is the Grashof number and Ra is the Rayleigh number, calculated by
(12)$$Ra_{L*}\;=\;Gr\cdot \mathrm{Pr}=\frac{g\cdot \beta_{a}\cdot \left(T_{s}-T_{0}\;\right)\cdot {L^{*}}^{3}}{\alpha \cdot \nu }$$
where *β_a_* is the coefficient of volumetric thermal expansion for ideal gas, *T_s_* is the sand mold surface temperature, *T_a_* is the atmosphere temperature and *g* is the gravitational constant.

The Nusselt number for convection in a closed air pocket also complies with Equation (10). The convection heat transfer is related to the thickness, width and their ratio of the air pocket, the temperature difference of the side walls and the orientation of the air pocket.

4. Radiation

Since the skeletal sand mold can reach a high temperature, the radiation among the irregular surface itself may be significant. To achieve accurate radiation heat transfer calculation, the view factor should be calculated. However, the calculation volume will increase. The view factor, which is between a pair of surfaces: a source surface and a target surface as shown in Figure 2, is calculated by the following equation.
(13)$$F_{v}=\frac{S_{1}S_{2}\cos\theta_{1}\cos\theta_{2}}{\pi r^{2}}$$
where *r* is the distance between the source and target surfaces, *S_1_* and *S_2’_* are the areas of the surfaces, $\theta_{1}$ and $\theta_{2}$ are the angles between the normal directions of the surfaces and their connection line, respectively.

### 3.2. Thermomechanical Coupled Modeling

The thermal analysis can provide the cooling history of the castings and the thermal history of the sand mold. To understand the evolution of residual stress and deformation, a stress analysis is necessary. A thermo-elasto-plastic model can be used for stress analysis. The temperature fields are used as thermal loads for stress analysis. The thermal expansion and contraction of the casting and mold lead to internal stress in both the casting and mold. The optimization of the shell thickness and support structure of the skeletal sand mold will be an important goal for modeling and simulation in order to conserve materials and manufacturing time of the sand mold by additive manufacturing and to avoid cracking or breaking in the meantime.

The constitutional equation is the relationship of the stress increment with strain increment and temperature increment.
(14)$$d\{\sigma \}=[\overset{-}{D^{ep}}]\left[d\{\varepsilon \}-\overset{-}{\beta }(T)dT\right]$$
where, {*σ*} is stress vector, including three main stress *σ_x_*, *σ_y_* and *σ_z_* and tangential stress *σ_xy_*, *σ_yz_* and *σ_xz_*, {*ε*} is the strain vector, including *ε**_x_*, *ε**_y_*, *ε**_z_*, *ε**_xy_*, *ε**_yz_* and *ε**_xz_* and *β* is the thermal expansion coefficient, a function of temperature. $\left[\overset{-}{D^{ep}}\right]$ is the elasto-plastic stiffness matrix.
(15)$$[\overset{-}{D^{ep}}]=\left[\overset{-}{\left[D^{e}\right]}-\frac{[\overset{-}{D^{e}}]\frac{\partial \overset{-}{\sigma }}{\partial \{\sigma \}}{\left\{\frac{\partial \overset{-}{\sigma }}{\partial \{\sigma \}}\right\}}^{T}\overset{-}{\left[D^{e}\right]}}{H^{\prime }+{\left\{\frac{\partial \overset{-}{\sigma }}{\partial \{\sigma \}}\right\}}^{T}[\overset{-}{D^{e}}]\frac{\partial \overset{-}{\sigma }}{\partial \{\sigma \}}}\right]$$
where $\bar{\sigma }$ is equivalent stress and *H’* is the hardening coefficient.

The material properties are related to the temperature and phase state. After solidification, some alloys undergo solid-phase transformation, which can greatly affect their mechanical properties. Kang et al. [20] considered the significant variation of mechanical properties of a kind of stainless steel that undergoes martensitic phase transformation during cooling in a numerical simulation of the deformation of a heavy hydro turbine blade casting.

#### 3.2.1. Treatment of Sand Mold

The mechanical behavior of the skeletal sand mold is very important in the thermomechanical coupled modeling and simulation. Its strength, measured by Shangguan et al. [21], declines significantly with temperature, as shown in Figure 3. Usually, the sand mold is neglected or treated as an elastic model. Xu et al. [18] treated the sand mold as a thermo-elasto-plastic model, but the simulation process was hard to converge. The skeletal sand mold will give less expansion resistance to the expansion or contraction of a casting; thus, it is beneficial to avoid cracks in a casting. The resin in the sand mold is burned off to some extent. The loose sand can keep its original position if there is still solid sand shell to back up it, but it cannot endure tensile force.

#### 3.2.2. Mechanical Boundary Condition

Kang et al. [22] stated that the casting/sand mold interface underwent very complicated deformation during the casting process. There will be slip, pressurized contact and air gap because of the expansion and contraction of the casting and sand mold as they cool down or are heated up. Xu et al. [18] adopted the contact elements method for the casting/sand mold boundary in stress analysis.

### 3.3. Numerical Simulation

The stress analysis of a casting process depends on the heat transfer results of the solidification and following cooling processes. Meanwhile, the variations in material properties and mechanical boundary conditions make the stress analysis itself very hard. A combination of the finite difference method and finite element method is usually used in thermal stress analyses for castings. Liu et al. [23] integrated the thermal analysis by the finite difference method and stress analysis by the finite element method for the simulation of heavy steel castings. Kang et al. [22] converted the finite difference meshes into finite elements so as to realize the thermal and stress analyses of the castings by the same meshes.

In the skeletal sand mold, regardless of the shell or the truss structure, no thick section is involved. Thus, it is necessary to adopt fine meshes for the enmeshment of castings and molds regardless of the finite difference method or finite element method. Finite difference meshes will lead to steps for inclined or curved surfaces, which will affect the results because the whole sand mold geometrical model is mainly of thin walls. Fine meshes require fine time increment during calculation in order to converge, which significantly increases calculation volume and time.

## 4. Case Study

### 4.1. Thermo-Mechanical Simulation of the Casting Process of a Stress-Frame Structure in Skeletal Sand Mold 

Stress frame structure with thick and thin bar is usually used for evaluation of different cooling features and stress evolution. Hereby, a stress frame casting specimen with bounding dimensions of 300 mm × 200 mm × 120 mm with a riser was adopted for analysis. It consists of one thick bar of 40 mm × 40 mm and two symmetrical thin bars of 20 mm × 8 mm linked with two beams at their ends, as shown in Figure 4a. The different contraction of the thick and thin bars would introduce stress into the casting during the casting process, and this was used to investigate its residual stress and deformation. A skeletal sand mold with a bounding size of 400 mm × 300 mm × 180 mm and a shell of 10 mm thickness as well as a lattice support structure with 10 mm × 10 mm bars and 50 mm × 50 mm × 50 mm spacing was designed using a software developed by Shuangguan et al. [14], as shown in Figure 4c. The casting and skeletal sand mold were meshed into tetrahedron elements, as shown in Figure 4b,d, and then the heat transfer during casting process was analyzed by ProCast, a commercial software from ESI Company for the casting analysis. The pouring process was neglected, and the initial temperature of the liquid metal was set as 720 °C. The meshed models of the casting and skeletal sand mold and their temperature fields at different time instants were transferred into ANSYS for stress analysis, with the temperature field results at different times during the casting process as the thermal loads. In ANSYS, element type solid 185 was used for mechanical analysis. The casting/mold boundary was found and treated as contact element model. The symmetrical xy, yz and xz planes passing through the casting and skeletal sand mold were set as constraint in z, x and y directions, respectively. An aluminum alloy A356 and furan resin-bonded silica sand mold were selected; their thermal and mechanical properties are shown in Figure 5 and Figure 6 and Table 1. The casting was treated as a thermo-elasto-plastic model, and the sand mold was treated as an elastic model. The heat transfer of the casting process with the skeletal sand mold by both natural cooling was simulated. The forced cooling was considered as air blowing of the skeletal sand mold from all directions. For comparison, the heat transfer and stress analysis of the casting process with a traditional sand mold was also conducted with the same sand mold profile as that of the skeletal sand mold.

### 4.2. Results and Discussions

#### 4.2.1. Temperature Fields of the Skeletal Sand Mold and Casting

The temperature fields of the casting and skeletal sand mold during the casting process are shown in Figure 7. The sand shell surface temperature increased to approximately 400 °C less than 500 s after pouring. The internal surface in contact with the casting reached and remained at 450 °C through almost the whole process until the temperature decrease of the casting. Before the shakeout at 4500 s, the sand shell was still above 200 °C, only 30 °C lower than the casting temperature. During the whole process, the support lattice of the skeletal sand mold stayed at room temperature.

The temperature fields of the casting and traditional sand mold are shown in Figure 8. It can be seen that the sand mold was heated from the contact layer to the far external surface. The internal layer was also heated to 450 °C, but its thickness was only a half of that of the skeletal sand mold. The external surface reached only approximately 100 °C after 5600 s when the shakeout temperature was reached. The casting in the traditional sand mold required 1200 s more than that by the new sand mold to reach shakeout temperature.

The comparison of the cooling curves of P1 of the castings under different sand molds extracted from the temperature fields is shown in Figure 9. They coincided during the solidification process. After the solidification process, the cooling of the casting in the skeletal sand mold was slower than that using the traditional sand mold because the cooling of the casting mainly relied on the heat absorption of the sand mold, and the traditional sand mold was thicker and possessed greater heat capability than the skeletal sand mold. However, as the casting decreased to 260 °C, the cooling of the casting in the skeletal sand mold became faster than that using the traditional sand mold. The reason is that the heat absorption capability of the traditional sand mold decreased significantly because its temperature approached that of the casting. However, the heat dissipation of the shell of the skeletal sand mold still retained a similar rate by convection and radiation to the environment because its external surface temperature was close to that of the casting. Thus, the cooling curves of the casting in the skeletal sand mold remained almost straight after the solidification platform, whereas its cooling curve in the traditional sand mold turned gradually flat. The shakeout of the casting using the skeletal sand mold was performed 1160 s earlier, or 21% shorter. The comparison of the temperature difference between thick and thin bars (P1 and P2) is shown in Figure 9b; it was greatly reduced from 185 to 120 °C using the skeletal sand mold. This decrease is beneficial for the reduction of residual stress because a more uniform temperature field was achieved.

The thermal histories of thick and thin bars (P1 and P2) and the corresponding area of skeletal sand mold and traditional mold (P3 and P4) are shown in Figure 10. Using the skeletal sand mold, the sand mold reached a peak of 500 °C and 350 °C for the thick and thin bar areas in less than 1000 s, respectively, and then the temperatures decreased as the cooling of the casting continued. However, the traditional sand mold slowly increased to 150 °C after 4500 s. Regardless of thick or thin bar areas, the skeletal sand mold cooling curves were almost parallel to those of the casting; only a very narrow temperature difference existed after the sand mold temperature reached a peak. Therefore, the resin binder in the internal surface of the skeletal sand mold would be burnt off by the melt.

The comparison of the heat flux distribution and directions for the skeletal sand mold and traditional sand mold during the casting process is shown in Figure 11. The heat flux of the skeletal sand mold was far greater than that of the traditional sand mold. The heat flux direction of the former was from the skeletal sand mold surface to the air, whereas for the latter the heat flux from the surface to the environment was not significant because of its lower surface temperature, as shown in Figure 12. In the traditional sand mold, the released heat of the casting was mainly absorbed by the sand mold. Since its size was far bigger than the casting and its weight was 9.6 times that of the casting, the cooling of the casting at the early stage was fast. However, as the sand mold was heated up, the temperature difference between the casting and sand mold surfaces became increasingly smaller, and the heat absorption became increasingly slower. In addition, the poor heat conduction in the sand mold also limited the heat transfer from inside to the external surface. However, for the skeletal sand mold the shell layer was heated up quickly, and its heat release to the environment mainly depended on the convection and radiation; hence, its heat dissipation rate was higher than for the traditional sand mold, especially during the later cooling stage. Furthermore, the skeletal sand mold had a 2.84 times bigger external surface than the traditional sand mold, which facilitated the heat release to the environment.

#### 4.2.2. Stress and Deformation of the Skeletal Sand Mold and Casting

The stress of the casting and mold during the casting process using the skeletal sand mold is shown in Figure 13.

The stress mainly evolved from the thin bars of the casting and then expanded to the whole casting, reaching a maximum of 80 MPa at the joints of the thin bars and beam. For the sand mold, the high-stress areas were mainly in the shell; the average level was approximately 2 MPa, and the highest value reached 12 MPa at the corner area corresponding to the beam and thick and thin bars. 

The stress *σ_y_* along the length direction of the bar is shown in Figure 14. There was tensile stress in the skeletal sand mold located at the outside corners corresponding to the joints of the ends of the thin bars and beam and compressive stress at the inside corner of the ends of the thin bars and beam. That means that the mold was susceptible to cracks at these areas. The stress in casting was mainly tensile stress concentrating on the thin bar area.

A comparison of both the von Mises stress and *σ_y_* of the casting using the skeletal and traditional sand molds is shown in Figure 15 and Figure 16. The stress in the former was less than that in the latter, and this was explained by the above mentioned results of the temperature difference decrease of the thick and thin bars using the skeletal sand mold.

The stress–time curve of the thick and thin bars (P1 and P2) is shown in Figure 17. Their stress using the skeletal sand mold was less than that using the traditional sand mold, and the stress of the thin bar was far higher than that of the thick bar. The stress increased with time. The stress using the skeletal sand mold reached 56 MPa in the thin bar and 7 MPa in the thick bar, and the stress using the traditional sand mold reached 67 MPa in the thick bar and 17 MPa in the thin bar.

The deformation of the skeletal sand mold and casting during the casting process is shown in Figure 18. 

This casting mainly contracted during the casting process, especially along the y direction. However, the thin bar bent outward, and the displacement was in the range of 0–1.3 mm. The skeletal sand mold underwent expansion at the beginning because of being heated up and then contraction at the middle parts resulted by the contraction of the casting. The displacement of the sand mold was in the range of 0–2.2 mm. The maximum displacement of the top of the sand mold reached 1.1 mm at the shakeout stage at the joint of the thin bars and beam.

A comparison of the final deformation of the casting using the skeletal and traditional sand molds is shown in Figure 19. The latter exhibited bigger deformation. Therefore, using the skeletal sand mold is beneficial for the reduction of residual stress and deformation of the casting.

#### 4.2.3. Effect of shell thickness on the stress and deformation of the casting and mold

The effect of the shell thickness (10 mm, 20 mm and 30 mm) of the skeletal sand mold on the cooling of the castings was studied. The comparison of the temperatures of the skeletal sand with different shell thickness corresponding to thick and thin bar areas (P3 and P4) is shown in Figure 20. The thicker the shell, the lower the temperature peak. The temperature peaks at P3 of the shell of thickness of 10 mm, 20 mm and 30 mm were 460 °C, 340 °C and 260 °C, respectively.

The cooling curves of the thick bar at P1 under different shell thicknesses are shown in Figure 21. Before 3500 s, the thicker the shell, the faster the cooling. After 3500 s, the thicker the shell, the slower the cooling. The temperature difference of thick and thin bars (P1 and P2) also varied with the shell thickness. The thicker the shell, the higher the temperature difference. As the shell became thicker and thicker, it was close to the traditional sand mold; hence, the cooling feature of the casting was similar to that using the traditional sand mold, and the residual stress would increase. 

The effect of shell thickness on the stress and deformation of the casting and skeletal sand mold is shown in Figure 22. As the shell thickness increased, the equivalent stress of the skeletal sand mold decreased: > 2 MPa for the 10 mm shell, 2 MPa for the 20 mm shell and < 2 MPa for the 30 mm shell. However, the stress of the casting changed in the opposite way, and the increase in the sand mold shell thickness led to an increase in residual stress.

The comparison of the deformation of the casting and skeletal sand mold is shown in Figure 23. The increase in shell thickness reduced the deformation of the sand mold, with the displacement decreasing from 0.9 for the 10-mm-thick shell to 0.2 mm for the 30-mm-thick shell, but it resulted in bigger deformation of the casting.

Thus, the shell thickness should be controlled to be as thin as possible to achieve less residual stress and deformation of a casting if the skeletal sand mold can endure the casting process without the appearance of cracks. Here, the 10-mm-thick shell is preferred.

#### 4.2.4. Improvement Design of the Skeletal Sand Mold

The stress result during the casting process on the skeletal sand mold is detrimental to the sand mold. The corners or junction areas are especially susceptible to cracking. The filleting in the sharp corners will be helpful for the avoidance of stress concentration. The comparison of the stress of the skeletal stand mold with and without filleting is shown in Figure 24. The stress of the mold was significantly reduced at some junction areas as marked in Figure 24a. The stress in the y direction was greatly reduced as well, as shown in Figure 25.

A comparison of the deformation of the casting and skeletal sand mold with and without filleting is shown in Figure 26. The deformation of the skeletal sand mold was also greatly reduced, while, the deformation of the casting was almost the same. Therefore, the filleting in the sharp corners of the skeletal sand mold is useful.

### 4.3. Validation

The above mentioned two skeletal sand molds, Mold 1 (without filleting) and Mold 2 (without filleting) and Mold 3 (Mold 1 without trusses at four corners) and one traditional sand mold were constructed using silica sand and furan resin as a binder by an ExOne-Smax 3D printing machine, as shown in Figure 27. During the printing process, a layer of loose sand mixed with a hardening agent was swept on the sand bed, and then the printer head containing an array of hundreds of tiny nozzles swept the whole sand bed and sprayed tiny drops of binder on the area corresponding to the sand mold. Then, sand sweeping and binder jetting were alternatively repeated, and finally the hollow sand molds were printed out. The sand was mainly quartz sand (99.1%) with a particle diameter of 75–150μm. The resin content was 1.6%–1.8%, and the curing agent was 0.2%. All of these molds were single-piece molds, i.e., no cope or drag. A356 Al alloy ingot was melted and then the melt was poured into these molds at room temperature with a pouring temperature of 710 °C. The traditional sand mold and skeletal sand molds was cooled by natural cooling. The temperatures of the castings and molds at P1–P4 were measured by thermocouples.

The geometric comparison of the designed skeletal sand molds and the traditional mold is listed in Table 2. It can be seen that at least 60% of the sand was saved by these new molds.

The experimental and simulated cooling curves of the stress frame specimen (P1 and P2) using the traditional sand mold and skeletal mold 1 are compared in Figure 28. It can be seen that simulated and experimental results were in good agreement. The cooling rate of the skeletal sand mold was faster than that of the traditional dense sand mold. Therefore, it was obvious that the skeletal sand mold could save cooling time of the casting.

The surface temperature of the sand mold was monitored using the infrared imaging camera Flir T250. A comparison is shown in Figure 29. The surface temperature of the sand mold was below 37 °C, and only a small area (where holes were drilled for the cleaning of unbonded sand in the mold and then sealed with clay) reached 97 °C. Thus, the heat released from the casting was mainly absorbed by the sand mold block, with less heat dissipation to the environment. The highest temperature zone in the middle was in the riser.

On the other hand, the surface temperature of the skeletal sand mold reached above 400 °C 250 s after pouring. Then, it decreased slightly to 320 °C 700 s later. Even after 2000 s, the shell temperature was still approximately 140 °C. The temperature of the shell surface was relatively uniform. The highest temperature was at the shell wrapping the thick bar. The lattice structure was at a lower temperature, approximately 80 °C; there was a steep decrease from the shell to the lattice. The high temperature of the shell led to heat dissipation to the environment by radiation and convection. It provides the potential for cooling enforcement by forced air or water spray cooling. 

The temperature variations of these points on the skeletal sand mold surface as shown in Figure 30a are shown in Figure 31. The shell was of high temperature while the lattice structure had a relatively lower temperature.

The deformation of the casting was obtained by the comparison of the measured profile using a laser scanning device (HSCAN made by Hangzhou Scantech Co., Ltd., China) and the original designed profile. The displacements of the castings using the skeletal and traditional sand mold are compared in Figure 32. Using the skeletal sand mold, the deformation of casting was reduced, validating the simulated results.

The conditions of these three skeletal sand molds are shown in Figure 33. Comparing mold 1 and mold 2 it can be seen that without filleting, cracks finally occurred in the joint of the bars, with filleting, no cracks were found. Comparing mold 1 and mold 3, four trusses at the four corners reduced some cracks. Therefore, the proper design of the skeletal sand mold with trusses at corners and filleting in the conjunctions is very important.

## 5. Conclusions

The modeling and simulation of the casting process using a skeletal sand mold was systemically analyzed. A numerical simulation of the casting process of a stress frame specimen using skeletal sand mold was performed. The temperature, stress and displacement fields of the casting and skeletal sand mold were obtained and compared with those of the traditional sand mold. Additionally, the simulated results were validated with experiments.
The shell of the skeletal sand mold was quickly heated to higher temperature than the traditional sand mold; thus, the dissipation of heat to the air can be significantly enhanced by convection and radiation. The binder in the internal layer of the sand shell would be burned off by the casting, which is a main consideration of the strength of the mold in the design of skeletal sand mold shell thickness.The skeletal sand mold is of complicated geometry, which leads to the hard selection of its external surface during the setting of boundary conditions. To achieve different cooling conditions for locations of interest of the skeletal sand mold, the selection of numerous local areas of the external surface is also a challenge in numerical simulation.The skeletal sand mold of the stress-frame casting reached 450 °C, and a very small temperature difference was kept with the casting. Its cooling power was almost the same during the whole cooling process; thus, it took 1200 s less time to reach shakeout temperature. Using the skeletal sand mold, the temperature difference between thick and thin bars could be reduced; thus, the residual stress and deformation of the stress frame casting could be decreased.The skeletal sand mold underwent significant stress and deformation, which made it susceptible to cracking. However, more trusses and filleting in the conjunctions of the skeletal sand mold could greatly reduce the stress and cracking tendency.A thicker shell led to the decrease of residual stress and deformation of sand mold, but the opposite results for the casting. Therefore, the shell thickness should be controlled to be as thin as possible to achieve less residual stress and deformation of a casting as if the skeletal sand mold could endure the casting process without the appearance of cracks.

## Figures and Tables

**Figure 1 materials-13-01596-f001:**
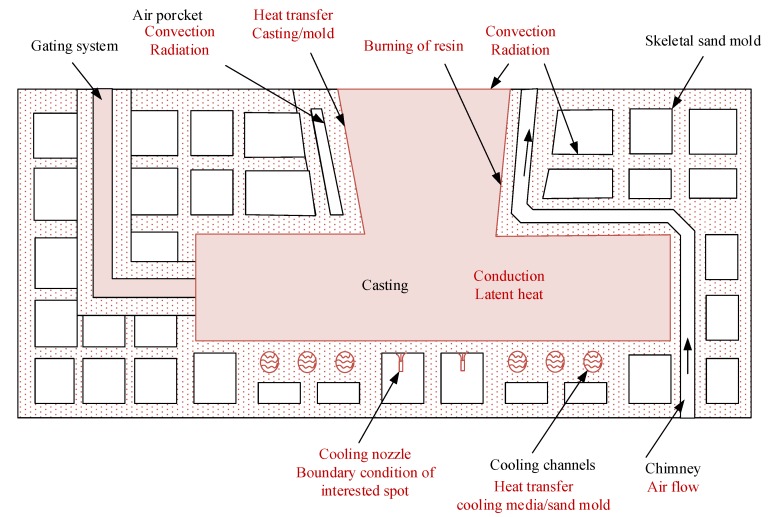
Heat transfer in the casting and skeletal sand mold.

**Figure 2 materials-13-01596-f002:**
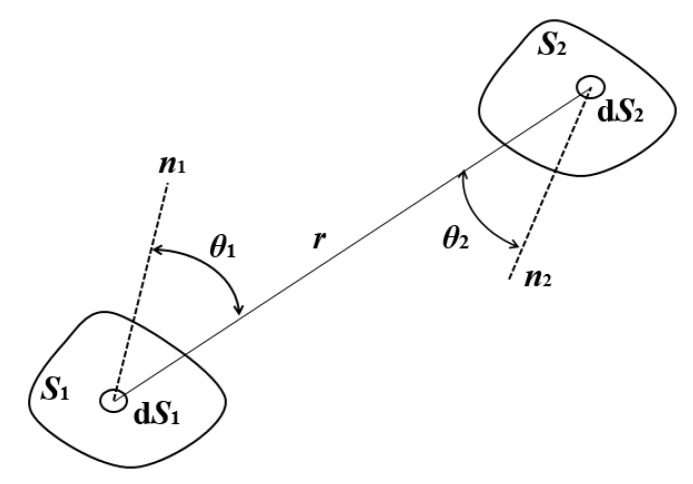
Illustrative diagram of view factor calculation between two areas.

**Figure 3 materials-13-01596-f003:**
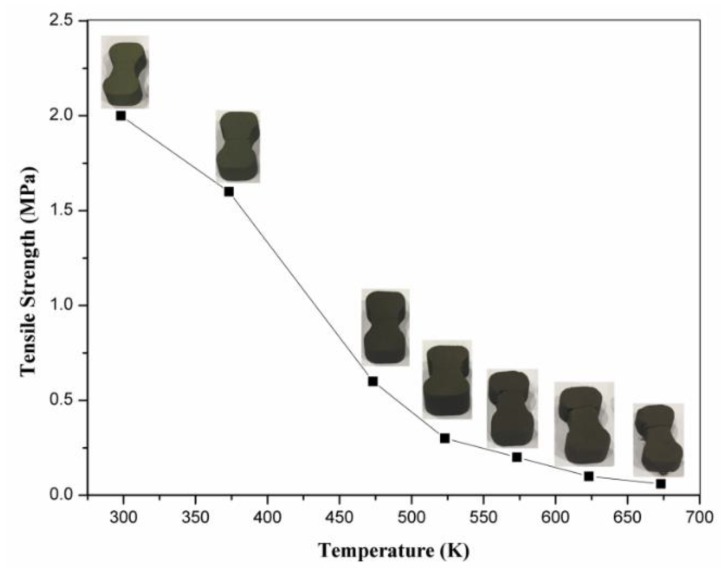
The relationship between sand mold strength and temperature.

**Figure 4 materials-13-01596-f004:**
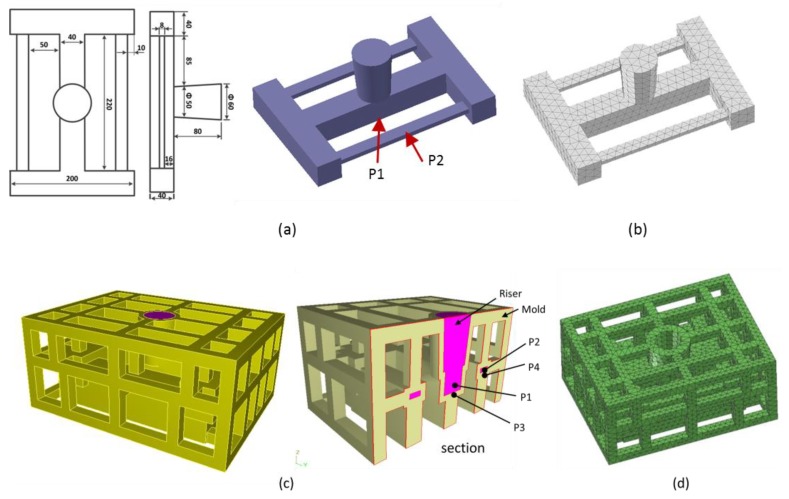
Casting and skeletal sand mold: (**a**) geometric model of the stress frame casting; (**b**) finite element model of the casting; (**c**) skeletal sand mold and (**d**) finite element model of the skeletal sand mold.

**Figure 5 materials-13-01596-f005:**
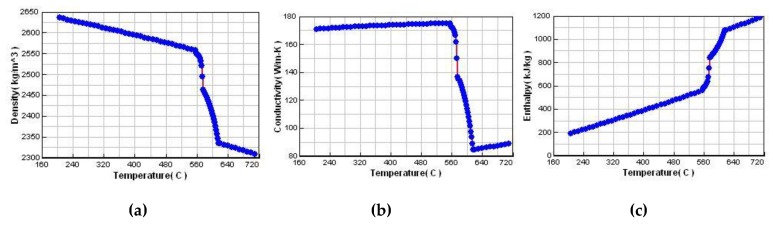
Thermal physical and mechanical properties of A356 alloy: (**a**) density; (**b**) heat conductivity; (**c**) enthalpy; (**d**) elastic modulus; (**e**) Poisson’s ratio and (**f**) thermal expansion coefficient.

**Figure 6 materials-13-01596-f006:**
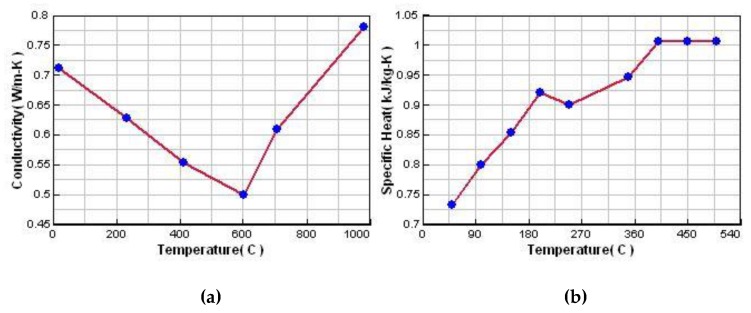
Thermal properties of sand mold: (**a**) thermal conductivity and (**b**) specific heat.

**Figure 7 materials-13-01596-f007:**
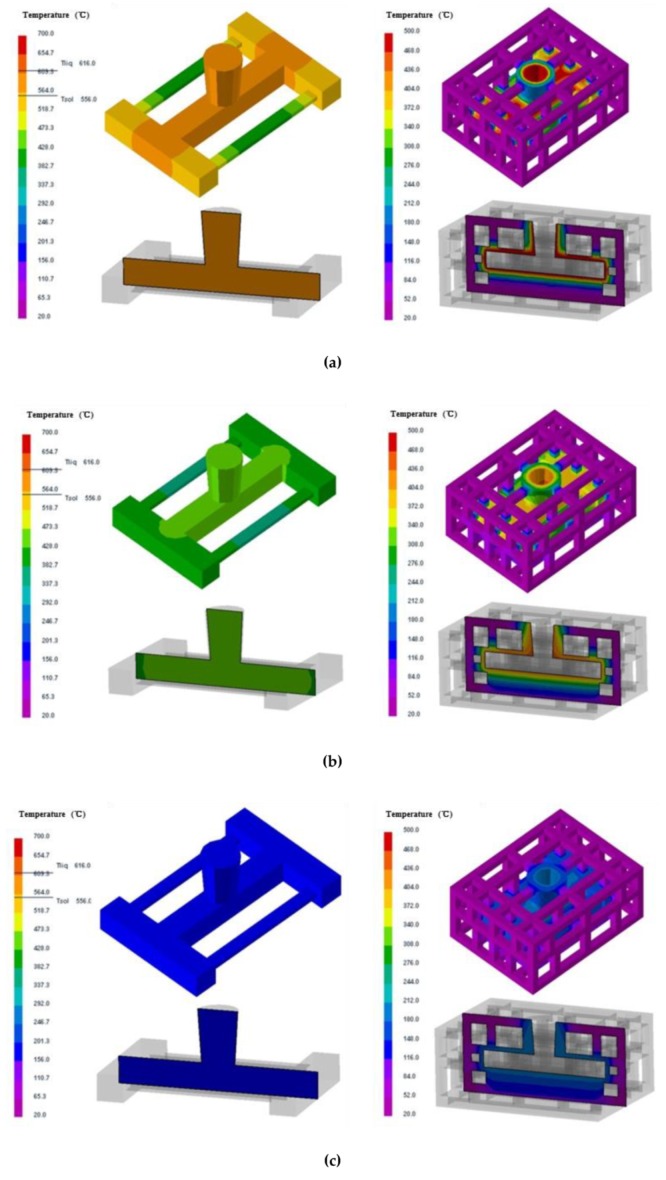
The temperature fields of the skeletal sand mold and casting: (**a**) 500 s; (**b**) 1500 s and (**c**) 4500 s.

**Figure 8 materials-13-01596-f008:**
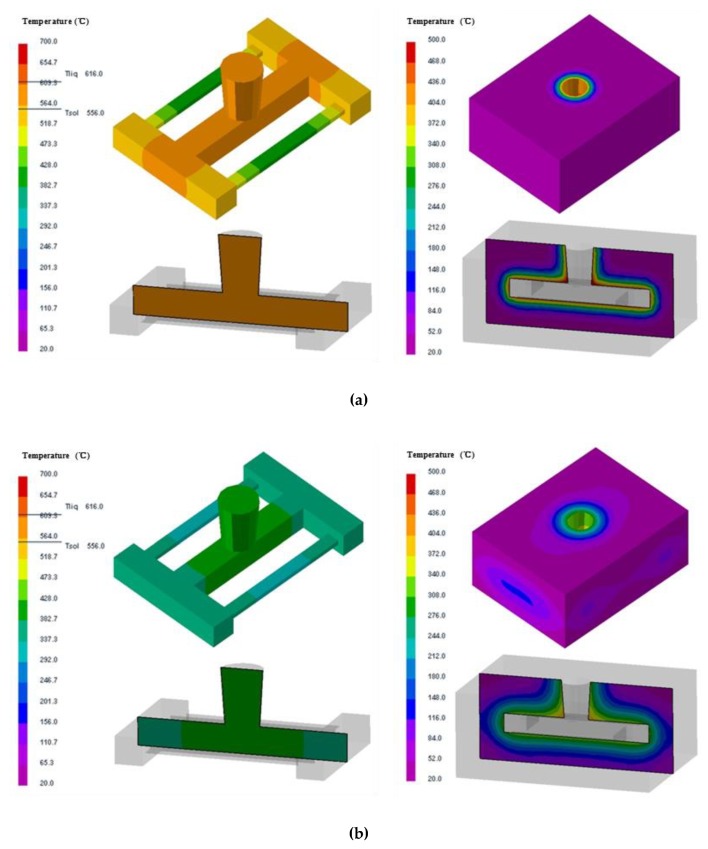
The temperature fields of the traditional sand mold and casting: (**a**) 500 s; (**b**) 1500 s and (**c**) 5600 s.

**Figure 9 materials-13-01596-f009:**
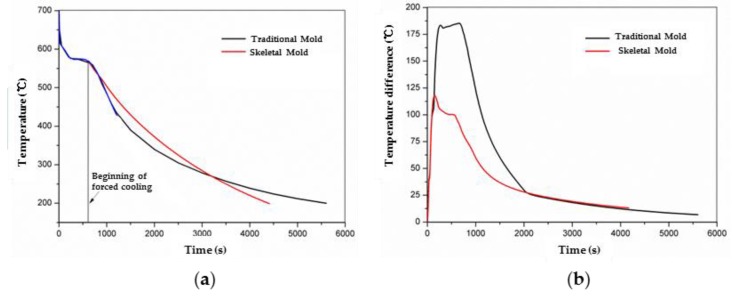
Comparison of cooling curves of castings in the skeletal and traditional molds: (**a**) cooling curves and (**b**) temperature difference of thick and thin bars

**Figure 10 materials-13-01596-f010:**
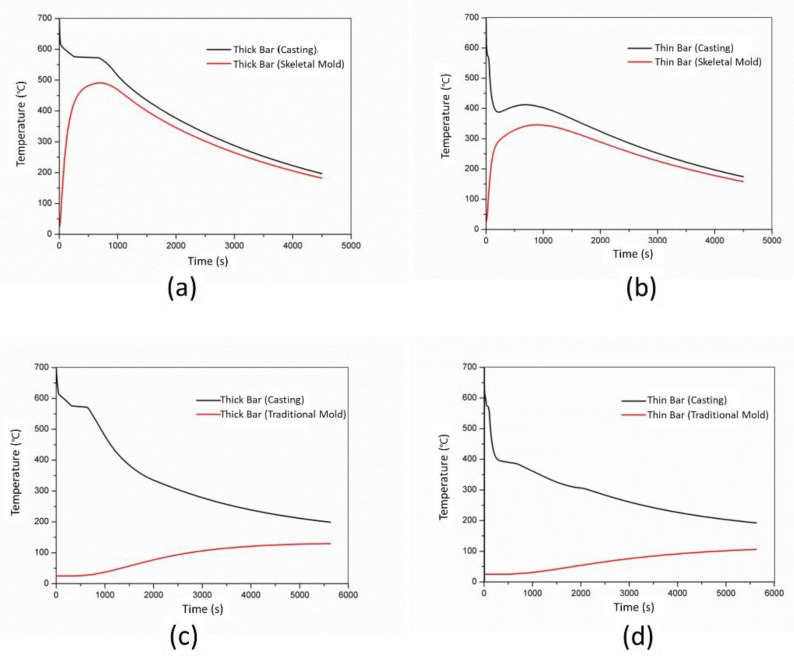
Comparison of the cooling and heating curves of the castings and sand molds: (**a**) thick bar with skeletal sand mold; (**b**) thin bar with skeletal sand mold; (**c**) thick bar with traditional sand mold and (**d**) thin bar with traditional sand mold

**Figure 11 materials-13-01596-f011:**
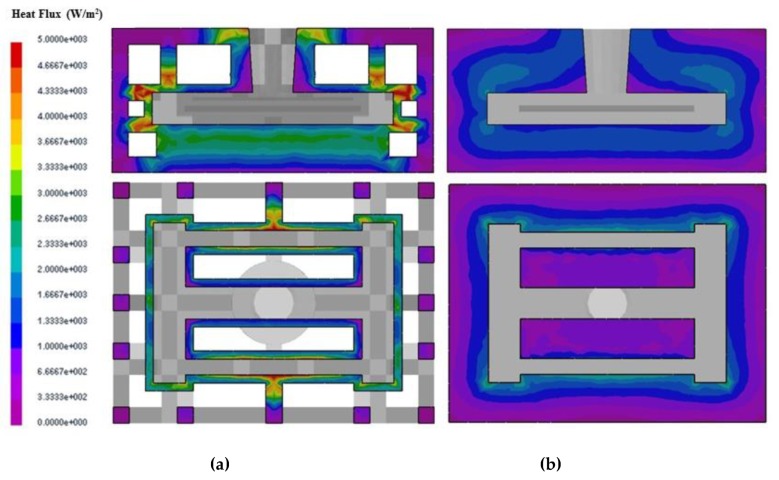
Comparison of heat flux of (**a**) skeletal and (**b**) traditional sand molds at 2500 s.

**Figure 12 materials-13-01596-f012:**
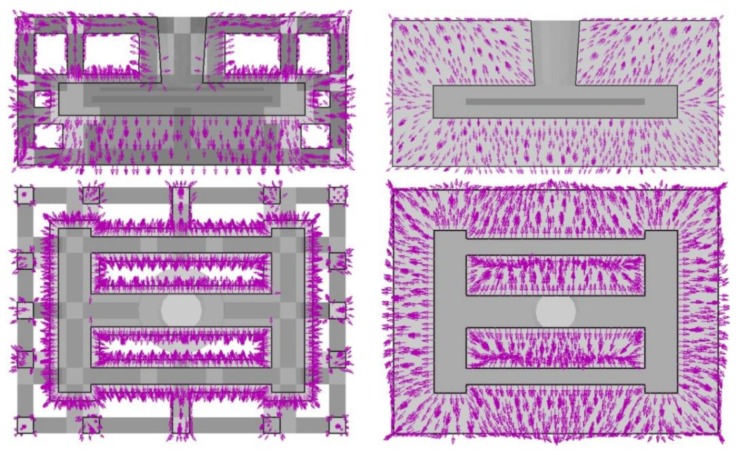
Comparison of the heat flux of (**a**) the skeletal sand mold and (**b**) traditional sand mold at 2500 s; the arrow size is proportional to the intensity of the heat flux.

**Figure 13 materials-13-01596-f013:**
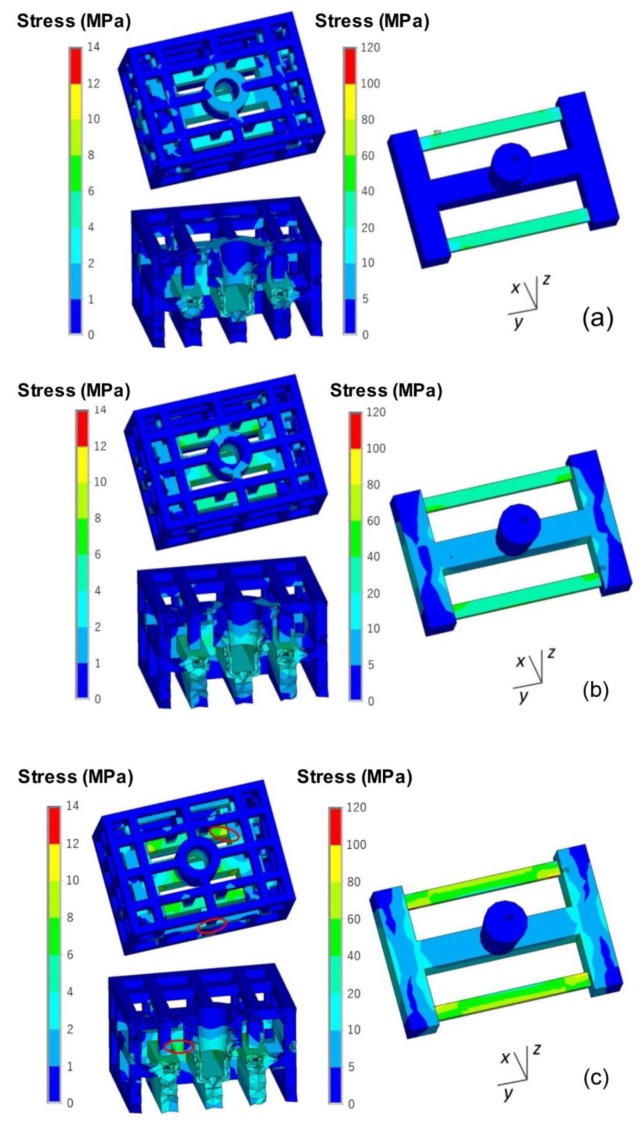
von Mises stress field of skeletal sand mold and casting: (**a**) 500 s; (**b**) 1500 s and (**c**) 4500 s.

**Figure 14 materials-13-01596-f014:**
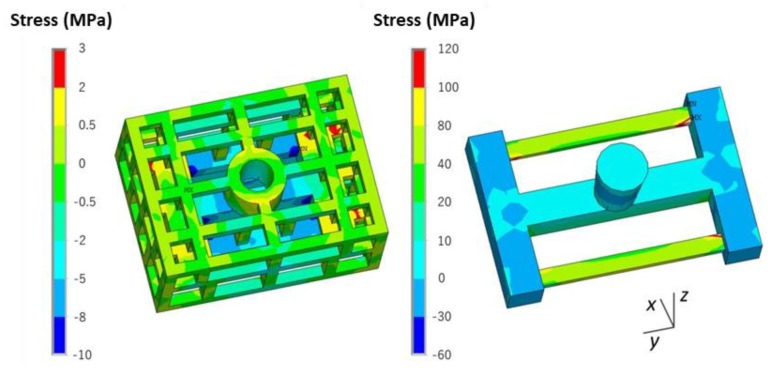
*σ_y_* distribution in the skeletal sand mold and casting at 4500 s.

**Figure 15 materials-13-01596-f015:**
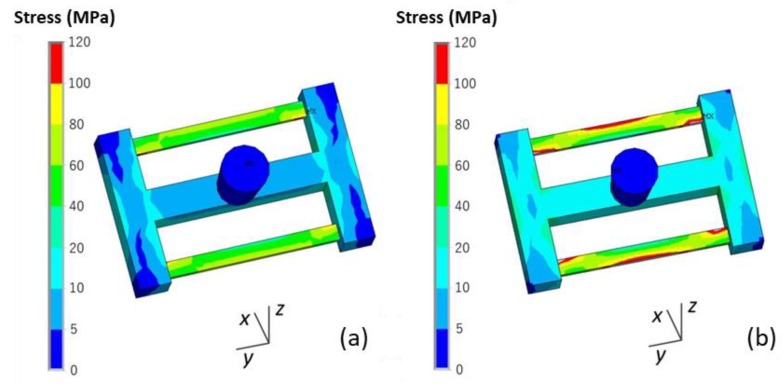
Comparison of the von Mises stress of the castings using the (**a**) skeletal sand mold and (**b**) traditional sand mold at shakeout.

**Figure 16 materials-13-01596-f016:**
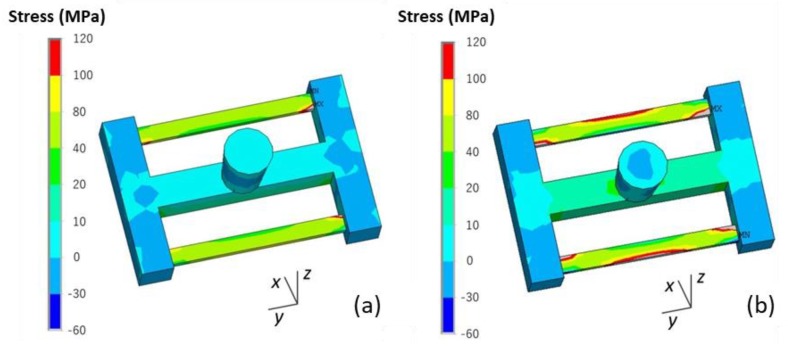
Comparison of the *σ_y_* of the castings using the (**a**) skeletal sand mold and (**b**) traditional sand mold at the shakeout.

**Figure 17 materials-13-01596-f017:**
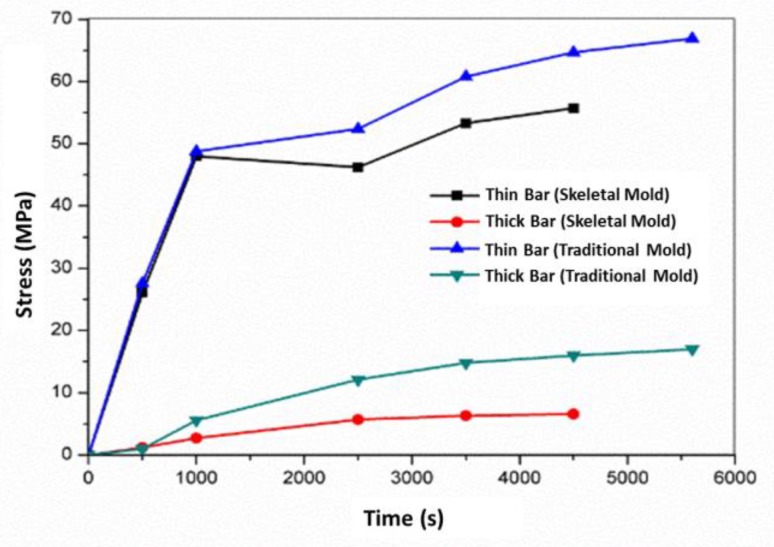
Stress evolution during the casting process in the casting.

**Figure 18 materials-13-01596-f018:**
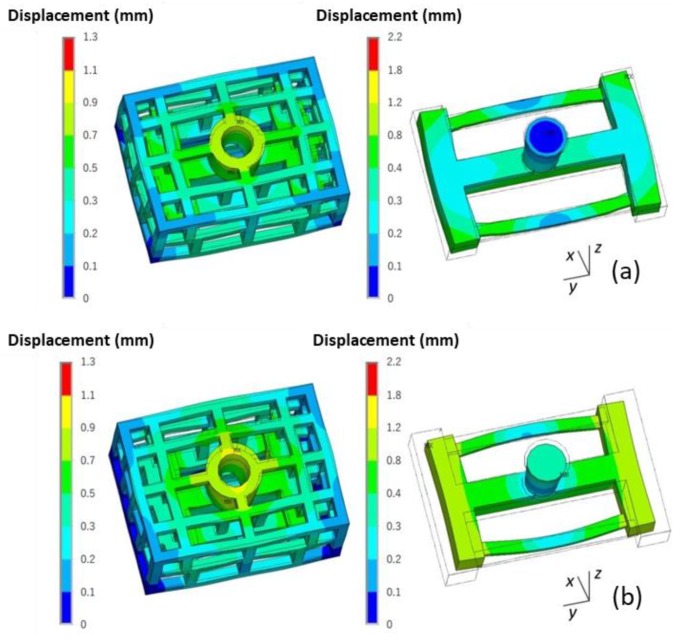
The deformation (equivalent displacement and deformed geometry) of the skeletal sand mold and casting (the wire frame is the original geometry, the rendered one is the final shape with magnified deformation, magnification ×20): (**a**) 500 s; (**b**) 1500 s and (**c**) 4500 s.

**Figure 19 materials-13-01596-f019:**
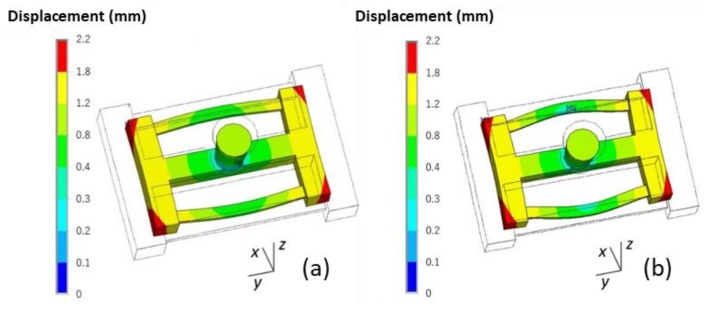
Comparison of the deformation (equivalent displacement and deformed geometry) of casting at shakeout using (**a**) the skeletal and (**b**) the traditional sand mold, magnification ×20.

**Figure 20 materials-13-01596-f020:**
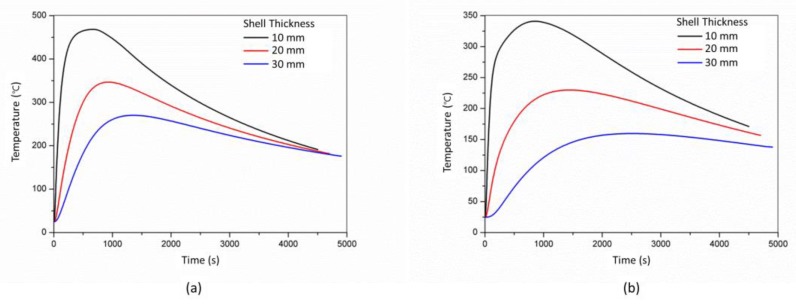
Comparison of cooling curves of (**a**) P3 and (**b**) P4 using the skeletal sand mold with a different shell thickness.

**Figure 21 materials-13-01596-f021:**
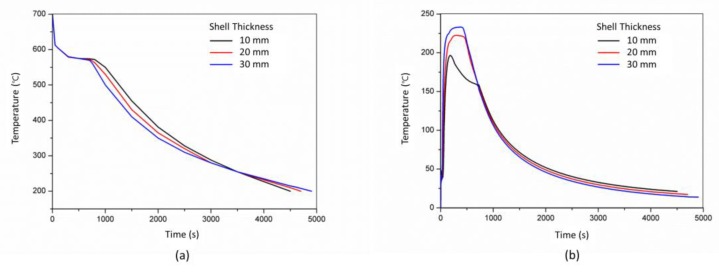
Comparison of cooling curves of P3 in the casting using the skeletal sand mold with different shell thicknesses: (**a**) cooling curves and (**b**) temperature difference between thick and thin bars (P1 and P2).

**Figure 22 materials-13-01596-f022:**
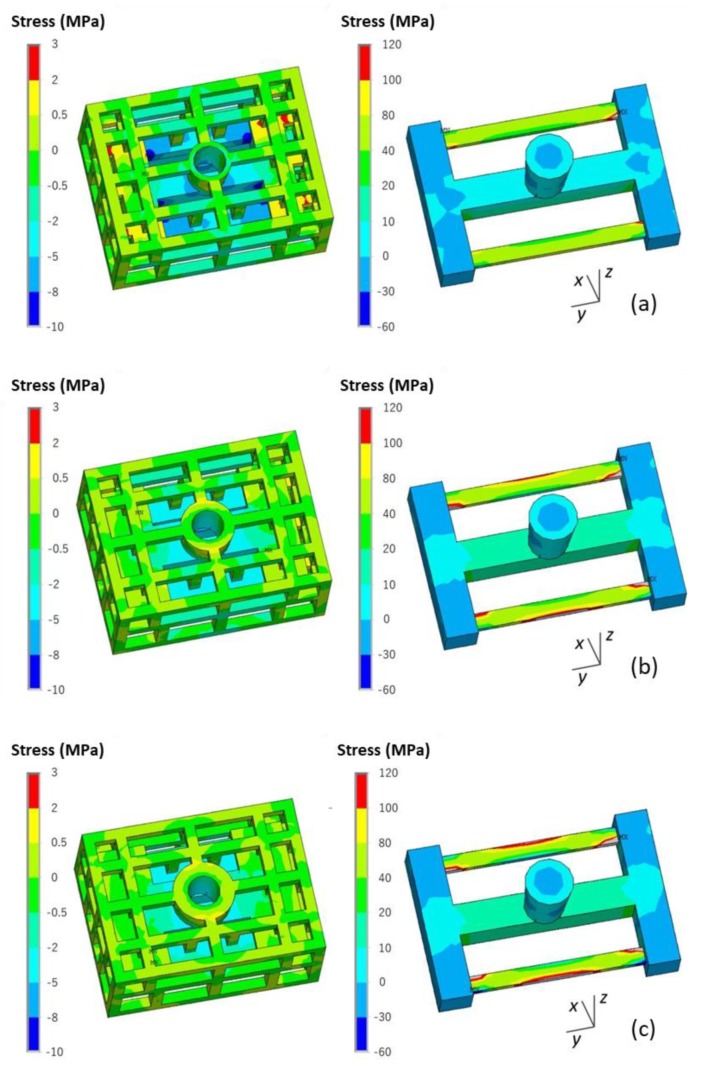
Comparison of the σ_y_ stress of the skeletal sand molds and castings using a skeletal sand mold of different shell thickness: (**a**) 10 mm; (**b**) 20 mm and (**c**) 30 mm.

**Figure 23 materials-13-01596-f023:**
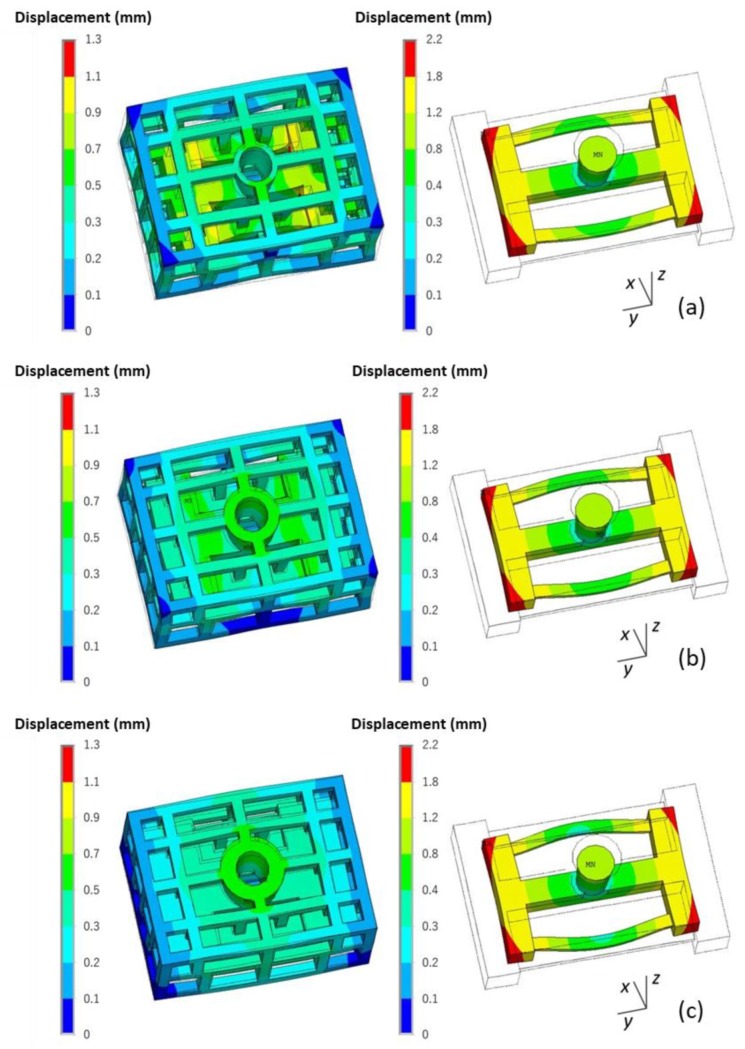
Comparison of the deformation (equivalent displacement and deformed geometry) of skeletal sand molds and castings using a skeletal sand mold of different shell thicknesses: (**a**) 10 mm; (**b**) 20 mm and (**c**) 30 mm

**Figure 24 materials-13-01596-f024:**
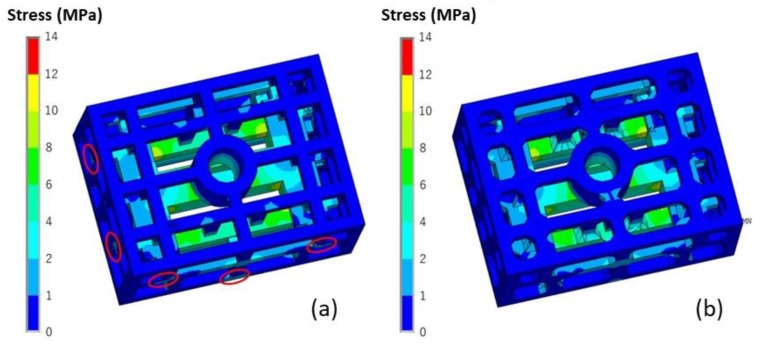
Comparison of the von Mises stress of the skeletal sand mold at shakeout (**a**) without filleting and (**b**) with filleting.

**Figure 25 materials-13-01596-f025:**
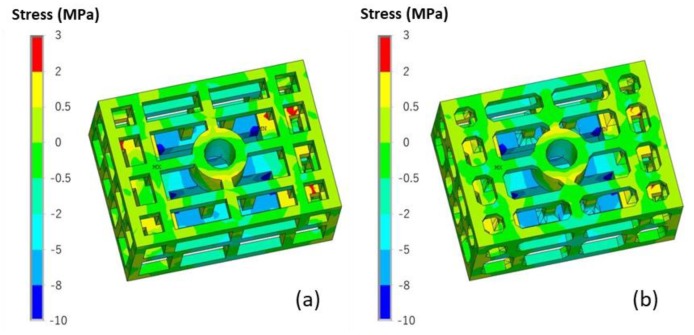
Comparison of the *σ_y_* stress of the skeletal sand mold at shakeout (**a**) without filleting and (**b**) with filleting.

**Figure 26 materials-13-01596-f026:**
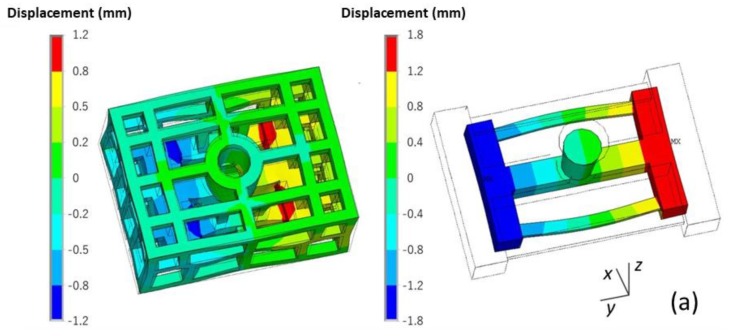
Comparison of the deformation (equivalent displacement and deformed geometry) of the skeletal sand mold at the shakeout (**a**) without filleting and (**b**) with filleting, magnification ×20.

**Figure 27 materials-13-01596-f027:**
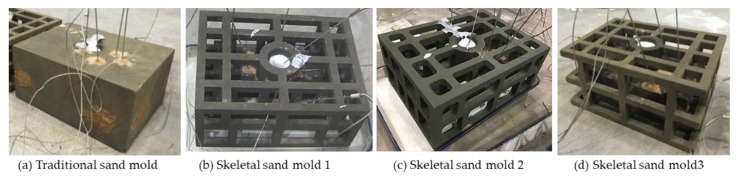
Onsite pouring experiments.

**Figure 28 materials-13-01596-f028:**
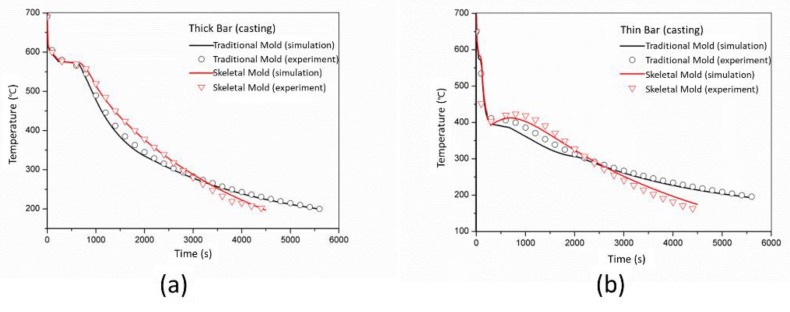
Comparison of the simulated and experimental cooling curves of thick (**a**) and thin (**b**) bars of the specimen

**Figure 29 materials-13-01596-f029:**
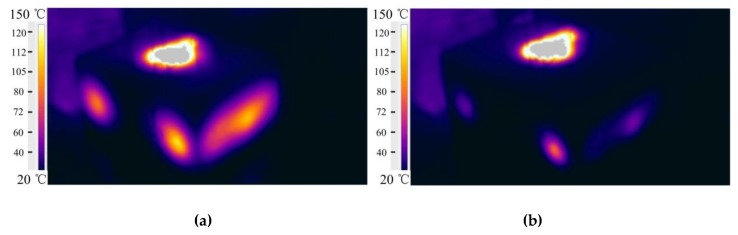
Infrared imaging of the traditional sand mold surface temperature: (**a**) 2000 s and (**b**) 5600 s.

**Figure 30 materials-13-01596-f030:**
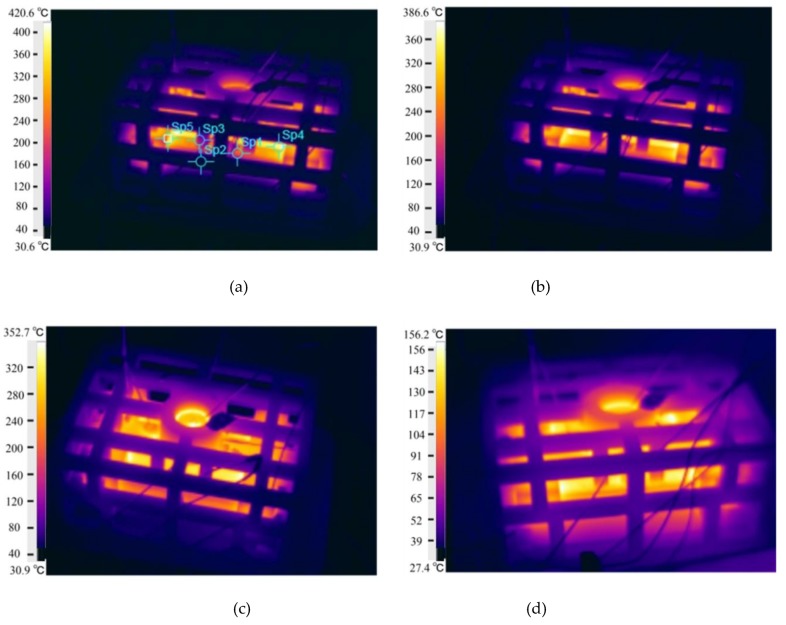
Infrared imaging of the skeletal sand mold surface temperature: (**a**) 250 s; (**b**) 300 s; (**c**) 700 s and (**d**) 2000 s.

**Figure 31 materials-13-01596-f031:**
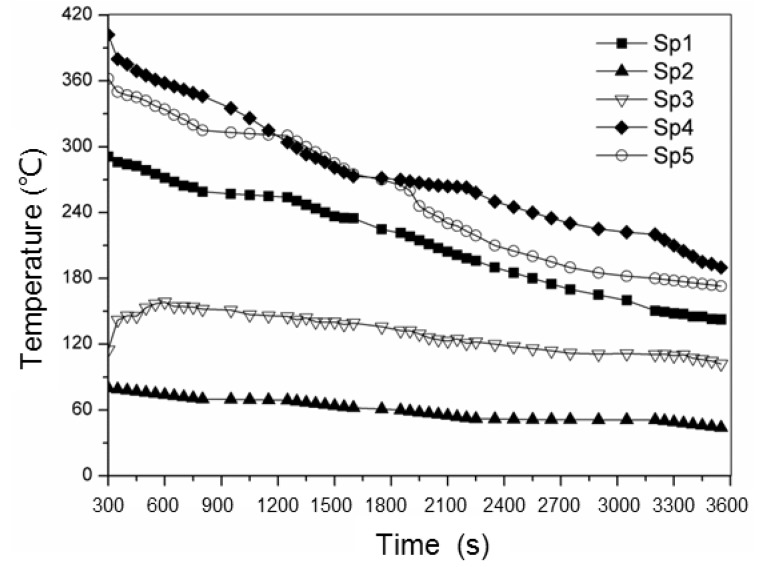
Cooling curves of the skeletal sand mold acquired by an infrared camera.

**Figure 32 materials-13-01596-f032:**
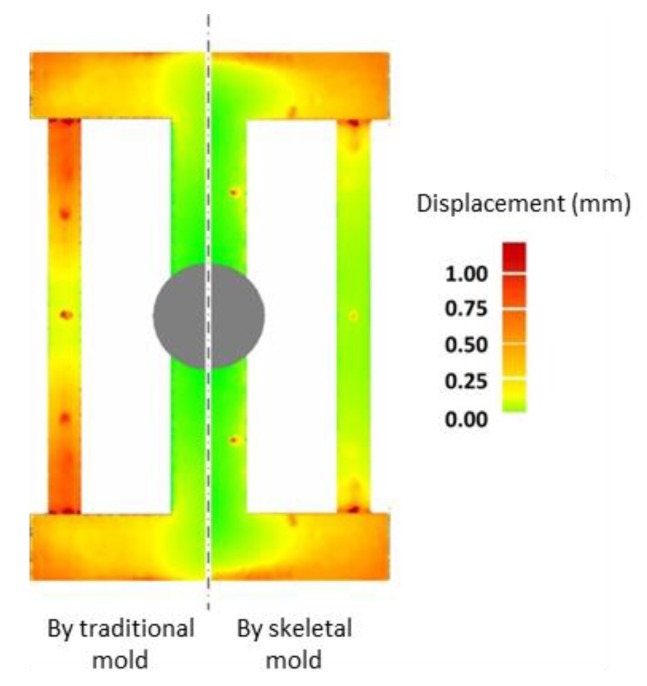
Comparison of deformation of the casting with traditional and skeletal sand molds.

**Figure 33 materials-13-01596-f033:**
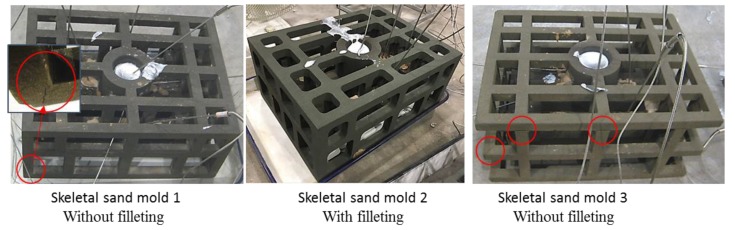
Comparison of cracks in the skeletal sand molds with and without filleting.

**Table 1 materials-13-01596-t001:** Other thermal and mechanical properties of A356 and sand mold.

Item	Parameter	Value
A356	Density (kg·m^−3^)	2680
	Liquidus (°C)	616
	Solidus (°C)	556
Sand mold	Density (kg·m^−3^)	1590
	Specific heat (Liang [24])	$C_{p}=0.9999+0.201\times 10^{-3}\times T+273.15-\frac{2.837\times 10^{4}}{{\left(T+273.15\right)}^{2}}$
	Elastic modulus (MPa)	5000
	Poisson’s ratio	0.3
	Thermal expansion coefficient (K^−1^)	1.33 × 10^−5^
	Emissivity	0.6
Casting/Sand mold	Boundary heat transfer (W·m^−2^·K)	616 °C 600 556 °C 250
Mold/Environment	Boundary heat transfer (W·m^−2^·K)	5

**Table 2 materials-13-01596-t002:** Comparison of skeletal molds and the traditional mold.

Type	Shell Thickness (mm)	Size of Lattice (mm)	Weight (kg)	Weight Reduction Rate (%)
Dense sand mold	none	none	30	0
Skeletal sand mold 1	10	20 × 20	10.2	66
Skeletal sand mold 2	10	20 × 20	10.2	66
Skeletal sand mold 3	10	20 × 20	10	66

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
