# Peer review of "Modeling and Simulation of the Casting Process with Skeletal Sand Mold"

_materials, 2020, doi:10.3390/ma13071596_

Round 1

Reviewer 1 Report

The present submission is a good contribution to the difficult topic on how the design of sand mold influences the cooling process of a casting and following cooling induced stresses.  The idea to optimize the thickness of the molding aggregate with respect to its influence on the cooling and stress induction is novel. 

The authors are kindly requested to clarify some minor uncertainties and consider the referes observation to the used references.

  1. The applicability of the optimization idea is strictly dependent on the knowledge related to the thermophysical and thermomechanical properties of the molding aggregates. 

A series of novel results were reported in the literature in this filed. It is regrettable that the authors have not considered a publication related to the exact same material as they have been used in their production of the molds.  Namely the furan resin bounded silica sand properties, produced by the ExOne 3D printing technique was investigated with respect to heat absorption capacity in the following publication https://doi.org/10.1007/s40962-016-0043-5  The authors are kindly requested to refer to this results in their work.

  1. The reported results based on numerical simulation are strictly dependent of the boundary heat transfer coefficients chosen in Table 1. It would be important to motivate why the authors chose these values.
  2. The authors indicate Liang [22] as a reference for the used specific heat in their work. On the other side in the reference chapter under position 22 the referred author is :Yingjiao L. Physical Chemistry. Metallurgy industry Press. 1983. Please clarify this mismatch.
  3. The authors use time dependent temperature variation curves in Figure 9,10,17 and 20. For the sake of clarity it would be correct to indicate more exactly where the reported cooling- or heating curves are taken from. (coordinates of the nodal points or similar description would improve the understanding).
  4. It ís difficult for the reader to observe the color scale for the local temperature observed by the infrared imaging method presented in Figure 30. Please modify the magnification of the color scale at least to the same size as in Figure 29.
  5. Figure 31 report the variation in the surface temperature observed by the infrared imaging method. All observed surface temperatures start with elevated temperatures, other than the ambient temperature and the authors indicate time 0. If we consider time 0 as the start of the casting or the end of the mold filling, then the initial temperature would be near to ambient with increase to a maximum during the casting cooling followed by a temperature reduction. Could the authors be kind and clarify the interpretation of this curves.

Good Luck!

Author Response

1) The applicability of the optimization idea is strictly dependent on the knowledge related to the thermophysical and thermomechanical properties of the molding aggregates. 

Answer:

Yes. The knowledge related to the thermophysical and thermomechanical properties of the molding aggregates is essential to the numerical simulation. The efforts to measure their properties have been done decades ago. Although the measured results are not in unanimous agreement, they are indeed helpful for numerical simulation. The properties used in this article are given in graphs and tables. For the progress of numerical simulation, the measurement of the thermal and mechanical properties of the printed sand mold is a meaningful research topic.

2) The reported results based on numerical simulation are strictly dependent of the boundary heat transfer coefficients chosen in Table 1. It would be important to motivate why the authors chose these values.

Answer:

The suggested paper about the investigation of degradation behavior of furan and phenolic resins in 3D printed sand cores is cited. These boundary conditions were widely used in the simulation researches

3) The authors indicate Liang [22] as a reference for the used specific heat in their work. On the other side in the reference chapter under position 22 the referred author is :Yingjiao L. Physical Chemistry. Metallurgy industry Press. 1983. Please clarify this mismatch.

Answer:

The reference is the same. In the reference list, the author’s first name and last name were reversed. The mistake “Yingjiao L.” is corrected as “Liang Y.”.

4) The authors use time dependent temperature variation curves in Figure 9,10,17 and 20. For the sake of clarity it would be correct to indicate more exactly where the reported cooling- or heating curves are taken from. (coordinates of the nodal points or similar description would improve the understanding).

Answer:

The curves in Figure 9,10,17 and 20 refer to the location of middle center of the thick and thin bars (P1 and P2, respectively) as marked in revised Fig.1.

5) It ís difficult for the reader to observe the color scale for the local temperature observed by the infrared imaging method presented in Figure 30. Please modify the magnification of the color scale at least to the same size as in Figure 29.

Answer:

The color scale in Fig.30 is corrected.

6) Figure 31 report the variation in the surface temperature observed by the infrared imaging method. All observed surface temperatures start with elevated temperatures, other than the ambient temperature and the authors indicate time 0. If we consider time 0 as the start of the casting or the end of the mold filling, then the initial temperature would be near to ambient with increase to a maximum during the casting cooling followed by a temperature reduction. Could the authors be kind and clarify the interpretation of this curves.

Answer:

The start time in Fig.30 is actually 300s. It is corrected.

Reviewer 2 Report

The paper “Modeling and simulation of casting process with skeletal sand mold” deals with the adoption of a skeletal sand mold for the casting process. Both numerical and experimental data have been presented in the paper and the final results are interesting.
But, in the opinion of this reviewer, the numerical analysis, which makes the present work original and different from the previous ones, should be strengthened by better explaining: (i) how the mechanical simulation was conducted in ANSYS; (ii) how the burning of resin and the convection were modelled in PROCAST (“convection heat transfer is related to the thickness, width and their ratio of the air pocket, the temperature difference of the side walls, and the orientation of the air pocket”); (iii) how the radiation model largely discussed by the authors in section 3.1.3 could be implemented in PROCAST.

The paper structure is requested to be improved: details about the simulated conditions and the experiments are not clear to the reader from the very beginning; in addition, experiments and simulated conditions need to be better correlated.

How the geometry of the case study was chosen should be explained and also supported by citing previous works in the literature.

The deformations are announced by the authors in figure 26 and 33, but in such figures displacements (mm) are shown.

The conclusion remark about the shell thickness of the mould (“A thicker shell leads to less stress”) should be better correlated to what the authors say about the effect of the shell thickness at lines 443-445 (“Thus, the shell thickness should be controlled to be as thin as possible to achieve less residual stress and deformation of a casting if the skeletal sand mold can endure the casting process without the appearance of cracks”).

Please correct the word density (now “Desnity”) in table 1.

Author Response

The paper “Modeling and simulation of casting process with skeletal sand mold” deals with the adoption of a skeletal sand mold for the casting process. Both numerical and experimental data have been presented in the paper and the final results are interesting. 
But, in the opinion of this reviewer, the numerical analysis, which makes the present work original and different from the previous ones, should be strengthened by better explaining: (i) how the mechanical simulation was conducted in ANSYS; (ii) how the burning of resin and the convection were modelled in PROCAST (“convection heat transfer is related to the thickness, width and their ratio of the air pocket, the temperature difference of the side walls, and the orientation of the air pocket”); (iii) how the radiation model largely discussed by the authors in section 3.1.3 could be implemented in PROCAST.

Answer:

The mechanical simulation conducted in ANSYS is given more details. The burning of resin can be predicted by the maximum temperature of the mold wall. The convection in the air pocket in ProCast is simplified as constant convection heat transfer coefficient in the case study. The “convection heat transfer is related to the thickness, width and their ratio of the air pocket, the temperature difference of the side walls, and the orientation of the air pocket” in section 3 aims at thoroughly discussing the heat transfer and stress evolution in the skeletal sand mold based casting process, actually, it poses a challenge for numerical simulation. Section 4 is a case study, the convection in air pockets is simplified because the complicated convection can’t be dealt with currently. Actually, in future, the numerical simulation should be improved for the skeletal sand mold based casting process.

The paper structure is requested to be improved: details about the simulated conditions and the experiments are not clear to the reader from the very beginning; in addition, experiments and simulated conditions need to be better correlated.

Answer:

Thanks for comments. This paper is mainly on the requirements of numerical simulation of this new skeletal mold design based casting process. So, firstly, this problem is thoroughly discussed on a broad view. Then a case study was given with experiment aimed for validation. The structure is emphasized in the introduction. “The new mold design brings a challenge to the modeling and simulation of the casting process developed for the traditional mold design. In the present study, the modeling and simulation of the casting process based on skeletal sand mold were investigated in a broad view, which would serve a guidance for the research of modeling and simulation corresponding to this kind of new mold design. Then, a case study about the simulation of a typical stress frame casting formed in a skeletal sand mold was analyzed to unveil their features of heat transfer, stress evolution and deformation.”

How the geometry of the case study was chosen should be explained and also supported by citing previous works in the literature.

Answer:

The choosing of the geometry of the case study is added. The stress frame shape casting is a kind of typical structure used for stress deformation of castings because of the difference of section thickness of the middle and side bars.

The deformations are announced by the authors in figure 26 and 33, but in such figures displacements (mm) are shown.

Answer:

The displacement is used to show deformation. In Fig.26, the wireframe is the original shape, which makes the deformed geometries clear. The explanation is added in the revised version.

The conclusion remark about the shell thickness of the mould (“A thicker shell leads to less stress”) should be better correlated to what the authors say about the effect of the shell thickness at lines 443-445 (“Thus, the shell thickness should be controlled to be as thin as possible to achieve less residual stress and deformation of a casting if the skeletal sand mold can endure the casting process without the appearance of cracks”).

Answer:

The effects of the shell thickness on the casting and sand mold are in opposite way. For sand mold, “A thicker shell leads to less stress”, however, “Thus, the shell thickness should be controlled to be as thin as possible to achieve less residual stress and deformation of a casting if the skeletal sand mold can endure the casting process without the appearance of cracks”. For better understanding, “A thicker shell leads to less stress” is changed as “A thicker shell leads to less stress in sand mold”.

Please correct the word density (now “Desnity”) in table 1.

Answer:

The spelling mistake is corrected.

Reviewer 3 Report

I had challenges in following the labeling scheme. It needs to be more consistent when referring to the two types of molds. For clarity, you should always refer to traditional sand mold as traditional sand mold.

Have you looked at other mold designs and other skeletal designs?

The abstract should be rewritten with quantified results.It reads very generic as-written.

Have you looked at differences in casting defects (i.e. oxide film inclusions and porosity)? You should include discussions on this in your paper.

Recent papers on 3DSP related to controlling casting performance has not been included.

I think the article is a little long. There are places where the wording could be more concise.

Author Response

Comments and Suggestions for Authors

I had challenges in following the labeling scheme. It needs to be more consistent when referring to the two types of molds. For clarity, you should always refer to traditional sand mold as traditional sand mold.

Answer:

Thanks for your comment, the traditional sand mold is used for the whole article.

Have you looked at other mold designs and other skeletal designs?

Answer:

In the introduction, the cooling channel buried mold, sand mold made by additive manufacturing and the skeletal sand mold design were mentioned. One more reference is added for the mold design based on 3D sand printing.

The abstract should be rewritten with quantified results.It reads very generic as-written.

Answer:

The abstract is improved.

Have you looked at differences in casting defects (i.e. oxide film inclusions and porosity)? You should include discussions on this in your paper.

Answer:

It is a good suggestion. This case study, stress frame casting design, is mainly aiming ot heat transfer and stress evolution features of the casting and new mold. Thus, porosity and microstructure based on the new mold design were not investigated in paper.

Recent papers on 3DSP related to controlling casting performance has not been included.

Answer:

Two more 3DSP articles are cited. Actually, referernces7~16 are related to 3DSP.

I think the article is a little long. There are places where the wording could be more concise.

Answer:

Yes. It seems a little long because it discussed the effects of serval parameters on heat transfer and stress evolution for the casting process basing on the new sand mold.

The paper is thoroughly checked again and wording is improved.

Round 2

Reviewer 2 Report

The authors have made some changes to the paper.

The structure of the paper is almost unchanged evenf if some clarifications useful to the reader have been included in the revised version.

The term deformation has not been changed and it is still adopted by the authors when commenting maps of displacements. It is clear that displacements give a idea about deofromation but, in the opinion of this Reviewer, it cannot be accepted to measure the deformation in mm (see for example lines 451-452).  

Author Response

Thanks for your comments.

The structure is modified as follows:

original

revised

4. Case Studies

5. Results and Discussions

5.1. Temperature fields of the skeletal sand mold and casting

5.2. Stress and deformation of the skeletal sand mold and casting

5.3. Effect of shell thickness on the stress and deformation of the casting and mold

5.4. Improvement design of the skeletal sand mold

6. Validation

7. Conclusions

4. Case Study

4.1. Thermo-mechanical simulation of the casting process of a stress-frame structure in skeletal sand mold

4.2 Results and Discussions

4.2.1 Temperature fields of the skeletal sand mold and casting

4.2.2 Stress and deformation of the skeletal sand mold and casting

4.2.3 Effect of shell thickness on the stress and deformation of the casting and mold

4.2.4 Improvement design of the skeletal sand mold

4.3 Validation

5. Conclusions

The “deformation” word are thoroughly checked, in lines 451-452 “The increase in shell thickness reduced the deformation of the sand mold from 0.9 mm for the 10-mm-thick shell to 0.2 mm for the 30-mm-thick shell” is changed as “The increase in shell thickness reduced the deformation of the sand mold, with the displacement decreasing from 0.9 mm for the 10-mm-thick shell to 0.2 mm for the 30-mm-thick shell”. In the captions of Figs.18, 19, 23 and 26, “Deformation” is revised as “Deformation (equivalent displacement and deformed geometry)”